# Depolarization and Hyperexcitability of Cortical Motor Neurons after Spinal Cord Injury Associates with Reduced HCN Channel Activity

**DOI:** 10.3390/ijms24054715

**Published:** 2023-03-01

**Authors:** Bruno Benedetti, Lara Bieler, Christina Erhardt-Kreutzer, Dominika Jakubecova, Ariane Benedetti, Maximilian Reisinger, Dominik Dannehl, Christian Thome, Maren Engelhardt, Sebastien Couillard-Despres

**Affiliations:** 1Institute of Experimental Neuroregeneration, Spinal Cord Injury and Tissue Regeneration Center Salzburg (SCI-TReCS), Paracelsus Medical University, 5020 Salzburg, Austria; 2Austrian Cluster of Tissue Regeneration, 1010 Vienna, Austria; 3Department of General, Visceral and Thoracic Surgery, University Clinic Salzburg, Paracelsus Medical University, 5020 Salzburg, Austria; 4Institute of Neuroanatomy, Mannheim Center for Translational Neuroscience (MCTN), Medical Faculty Mannheim, Heidelberg University, 68167 Mannheim, Germany; 5Department for Womens’ Health, Tübingen University, 72076 Tübingen, Germany; 6Institute of Physiology and Pathophysiology, Medical Faculty Hospital, Heidelberg University, 69120 Heidelberg, Germany; 7Institute for Anatomy and Cell Biology, Johannes Kepler University Linz, Krankenhausstrasse 5, 4020 Linz, Austria; 8Institute for Stem Cell Biology and Regenerative Medicine, Stanford University, Stanford, CA 94305, USA

**Keywords:** spinal cord injury, axotomy, primary motor cortex, corticospinal tract, HCN channels, I_h_ current

## Abstract

A spinal cord injury (SCI) damages the axonal projections of neurons residing in the neocortex. This axotomy changes cortical excitability and results in dysfunctional activity and output of infragranular cortical layers. Thus, addressing cortical pathophysiology after SCI will be instrumental in promoting recovery. However, the cellular and molecular mechanisms of cortical dysfunction after SCI are poorly resolved. In this study, we determined that the principal neurons of the primary motor cortex layer V (M1LV), those suffering from axotomy upon SCI, become hyperexcitable following injury. Therefore, we questioned the role of hyperpolarization cyclic nucleotide gated channels (HCN channels) in this context. Patch clamp experiments on axotomized M1LV neurons and acute pharmacological manipulation of HCN channels allowed us to resolve a dysfunctional mechanism controlling intrinsic neuronal excitability one week after SCI. Some axotomized M1LV neurons became excessively depolarized. In those cells, the HCN channels were less active and less relevant to control neuronal excitability because the membrane potential exceeded the window of HCN channel activation. Care should be taken when manipulating HCN channels pharmacologically after SCI. Even though the dysfunction of HCN channels partakes in the pathophysiology of axotomized M1LV neurons, their dysfunctional contribution varies remarkably between neurons and combines with other pathophysiological mechanisms.

## 1. Introduction

Axotomy causes structural and functional remodeling of corticospinal neurons and of the surrounding network. On the one hand, retrograde neuromodulation causes synaptic rearrangement and affects neurotransmission; on the other, the intrinsic excitability and output of axotomized neurons undergo chronic changes [1]. These events contribute to the neocortical pathophysiology after SCI [2], where remodeling involves altered excitability, altered inhibition, and altered cortical output [3,4,5,6,7,8]. Nevertheless, the pathophysiological role of axotomized corticospinal neurons after SCI is resolved only to a limited extent [9,10,11]. Thus, exploring the mechanisms of cortical imbalance is crucial to better understand, control, and support the process of regeneration after SCI [5,12,13].

In this study, we investigated the consequences of axotomy on the function of principal neurons of the primary motor cortex layer V (M1LV). In a rat model of SCI, transection of the dorsal corticospinal tract (CST) was carried out at the cervical level. This injury causes limited secondary damage and limited locomotor impairment [14,15]. Such conditions are ideal to highlight the consequences of axotomy for cortical motor neurons with minimal interference by inflammatory processes. In this model, we performed our analyses one week after injury, i.e., during the subacute SCI phase. At this time, the altered cortical activity reflects the increased excitability of axotomized cortical neurons [1,16,17]. At the same time, chronic symptoms and cortical inflammation have not manifested yet [18].

Hyperpolarization cyclic nucleotide gated channels (HCN channels) modulate neuronal functions in multiple ways. First, HCN channels open upon membrane hyperpolarization and contribute to neuronal depolarization via a mixed cationic conductance. Second, open HCN channels contribute to low input resistance (R_in_), thus increasing the input current necessary to elicit action potential (AP) firing in neurons. Third, the gating properties of HCN channels depend on intracellular concentration of cyclic nucleotides [19]. Accordingly, dysregulation of HCN channels can occur due to heterogeneous causes (e.g., chronic changes in membrane potential or in intracellular concentration of cyclic nucleotides) and cause various pathophysiological phenotypes associated with neurological disorders of the central and peripheral nervous system [20].

Here, we hypothesized that HCN channels are involved in the processes of cortical pathophysiology after SCI and lead to increased intrinsic excitability of axotomized cortical neurons. Furthermore, we reasoned that these channels may represent suitable targets to fine-tune the excitability of cortical networks. For this reason, we explored the role of HCN channels in relation to dysfunction of M1LV neurons after SCI. Moreover, we questioned whether these channels could be targeted by pharmacological treatments aiming to control neuronal excitability and output.

## 2. Results

Distal axotomy alters plasticity and increases excitability of infragranular neocortical layers [1]. To explore the cellular mechanisms behind such pathological changes, we examined the excitability of M1LV neurons in acute brain slices. We compared the function of uninjured neurons in healthy rats to the function of axotomized neurons in rats that underwent SCI. The population of principal neurons of M1LV was readily recognizable based on the typical large pyramidal morphology of the soma. Moreover, after SCI, axotomized neurons were marked by the retrograde tracer fluorogold (FG). Therefore, FG labeling identified the neurons as axotomized. Single cells were dialyzed with biocytin during patch clamp experiments (via patch pipette) and labeled with streptavidin during the histological analyses, which, in turn, allowed us to identify the FG+ axotomized neurons targeted during patch-clamp recordings (Figure 1A).

Patch-clamp experiments on uninjured neurons and on axotomized neurons allowed us to examine patterns of AP firing and to determine whether axotomy changed in the intrinsic M1LV neuron excitability (Figure 1B). Uninjured neurons had a resting membrane potential (RMP) of −72.9 ± 5.0 mV. Conversely, the RMP of axotomized neurons was significantly depolarized (−68.4 ± 5.1 mV, *p* = 0.018; Figure 1C and Table 1). Moreover, the rheobase of uninjured neurons was equal to 102.3 ± 43.7 pA. By comparison, the rheobase of axotomized neurons was significantly lower, i.e., equal to 59.3 ± 21.2 pA (*p* = 0.03; Figure 1D and Table 1). A lower rheobase implies that smaller input suffices to trigger AP firing in axotomized neurons. Thus, the excitability of M1LV neurons was increased after SCI. Apart from altered RMP and rheobase, the other electrophysiological parameters of axotomized neurons were comparable to those of healthy controls (Figure 1E–H and Table 1).

HCN channels play a prominent role in controlling the physiology of corticospinal neurons [21]. Additionally, these channels are involved in controlling both the RMP and the rheobase [19]. Thus, we questioned whether altered conductance mediated by HCN channels (I_h_) was involved in the functional phenotype of the axotomized M1LV neurons. For this reason, we analyzed the I_h_ current in both uninjured and axotomized M1LV neurons. Hyperpolarizing voltage steps elicited I_h_ currents endowed with typical slow activation kinetics and current tails (Figure 2A,B). These currents were readily blocked by the selective HCN channel blocker ZD7288 (100 µM). The maximal I_h_ current amplitude (Figure 2C and Table 1) and the voltage of I_h_ half-maximal activation (V_half,_ Figure 2D,E and Table 1), were comparable between uninjured and axotomized neurons. Strikingly, in uninjured neurons, the maximal I_h_ current amplitude correlated with the RMP depolarization. In contrast, in axotomized neurons, such a correlation did not occur (Figure 2F and Table 1). Possibly, this lack of correlation resulted from the small I_h_ currents in a subgroup of injured neurons, whose RMP was most depolarized (≥−66 mV; highlighted in Figure 2F,G). Hence, we hypothesized that chronic RMP depolarization may limit the functional relevance of HCN channels in some axotomized neurons.

To explore the matter, we questioned whether HCN channel activation contributed to RMP depolarization and whether the I_h_ current blockage would rescue axotomized neurons from depolarization. To this end, the HCN channel blocker ZD7288 was applied during current clamp experiments. We predicted that the effects of HCN channel blockage would be less prominent in cells depolarized beyond the voltage range of HCN channel activation (Figure 2E). In presence of ZD7288, the RMP of uninjured neurons and axotomized neurons were no longer significantly different (*p* = 0.23; Figure 3A,B). Additionally, a detailed analysis revealed different drug effects in uninjured and in axotomized neurons. After HCN channel blockage, uninjured neurons underwent RMP hyperpolarization uniformly (*p* < 0.001; Figure 3C and Table 2).

Conversely, in axotomized neurons, the effects of ZD7288 depended on whether the RMP was below −66 mV (SCI_HP_, HP = hyperpolarized neuron) or above −66 mV (SCI_DP_, DP = depolarized neuron). On the one hand, ZD7288 caused significant hyperpolarization in SCI_HP_ neurons (*p* < 0.0001); on the other hand, the drug failed to hyperpolarize SCI_DP_ neurons and caused a slight depolarization instead (*p* = 0.0051; Figure 3D and Table 3). 

The ionic conductance across the cell membrane is inversely proportional to the R_in_. For this reason, a decrease in ionic conductance across the cell membrane causes an increase in R_in_. Accordingly, the blockage of ionic conductance upon ZD7288 application was expected to cause an increase in R_in_. Indeed, ZD7288 increased the R_in_ of uninjured neurons (*p* < 0.001), and it increased the R_in_ of most of axotomized neurons (Figure 3E–G and Table 2 and Table 3). However, the effects of HCN blockage on R_in_ were more prominent and significant in SCI_HP_ neurons (*p* < 0.0001). Conversely, in SCI_DP_, changes in R_in_ did not reach significance (*p* = 0.3868). Note that R_in_ was measured with voltage steps between −70 mV and −75 mV, which allowed HCN channel activation in all neurons, regardless of their RMP.

According to Ohm’s law, the R_in_ relates to the rheobase inversely. Thus, application of ZD7288 was expected to decrease the rheobase as a consequence of the increasing R_in_ (Figure 4A, pink line). At the same time, since a larger input is necessary to trigger AP firing when the RMP is hyperpolarized, ZD7288 was also expected to increase the rheobase as a consequence of hyperpolarization (Figure 4A, green line). Therefore, we predicted that ZD7288 would have dual and opposing effects on the rheobase of M1LV neurons due to its simultaneous effects on R_in_ and RMP, one of which would necessarily predominate. At the same time, heterogeneous I_h_ amplitude and different extents of RMP depolarization hindered a precise prediction about which of the two effects would be the most predominant.

To resolve this matter, we measured the effects of ZD7288 directly, analyzing the rheobase changes in uninjured and in axotomized neurons upon drug application. Uninjured neurons responded to ZD7288 with a significant increase in rheobase (*p* = 0.03; Figure 4B,C and Table 3). On the other hand, SCI_HP_ and SCI_DP_ neurons responded to ZD7288 differently. Upon drug application, the rheobase of SCI_HP_ neurons increased significantly (*p* = 0.0391), but the rheobase of SCI_DP_ neurons did not increase (Figure 4B,D and Table 3). These data imply that the impact of ZD7288 in uninjured neurons and in SCI_HP_ neurons relied on RMP hyperpolarization predominantly and that the R_in_-mediated effects were minor. Notably, the application of ZD7288 did not elicit RMP hyperpolarization in SCI_DP_ neurons (Figure 3), and this likely explains why the drug also failed to increase their rheobase.

To highlight the R_in_-mediated effects and minimize the impact of RMP modulation by ZD7288, we measured the rheobase from an equalized baseline voltage of −70 mV (rheobase_-70_), i.e., a voltage that approximated the average RMP of uninjured neurons. In these conditions, we observed a low rheobase_-70_ in uninjured neurons and in injured neurons alike (Figure 4E and Table 4).

The rheobase of uninjured neurons increased by 20 ± 27 pA upon ZD7288 application (unequalized RMP). Conversely, their rheobase_-70_ decreased by 42 ± 32 pA. After drug application, the rheobase of SCI_HP_ neurons increased by 16 ± 23 pA. For the opposite situation, their rheobase_-70_ decreased by 25 ± 23 pA. The depolarization of SCI_DP_ neurons hindered a precise equalized-rheobase measurement due to relative instability of the membrane potential upon ZD7288 application and sparse spontaneous AP firing. However, in SCI_DP_ neurons devoid of spontaneous AP firing (*n* = 3), the rheobase decreased by 15 ± 22 pA upon ZD7288 application. Similarly, their rheobase_-70_ decreased by 3 ± 23 pA. Interestingly, in presence of ZD7288, the rheobase_-70_ of uninjured neurons was significantly higher than that of SCI_HP_ neurons (*p* = 0.03, Figure 4F, Table 4). Thus, regardless of HCN channel activation and in absence of chronic depolarization, axotomized neurons were more excitable than uninjured neurons.

## 3. Discussion

SCI damages corticospinal neurons and alters cortical excitability and output [5,22,23,24,25,26,27,28]. The axotomy caused by trauma, as well as the following inflammatory process and metabolic stress, may change HCN channel activity in injured neurons [29,30,31]. In turn, dysfunctional HCN channels are known to affect the neuronal RMP and intrinsic excitability [19]. Our recordings in axotomized M1LV neurons revealed altered RMP and rheobase after SCI. For this reason, we investigated the involvement of HCN channels in the pathophysiology of M1LV neurons after SCI. Furthermore, we considered whether the modulation of HCN channels after SCI may help to control the process of functional network remodeling, as already shown in other pathological models [20]. Thus, we questioned whether HCN channels were suitable targets for a pharmacological treatment.

To explore the involvement of HCN channels in cortical pathophysiology, we analyzed the effect of pharmacological HCN channel modulation on the excitability axotomized M1LV neurons in a model of SCI one week after injury. In this system, our analysis revealed that SCI did not affect the intrinsic properties of HCN channels in M1LV neurons per se. However, a striking pathophysiological feature after SCI was the chronic RMP depolarization in some neurons, which reached values beyond the voltage range of HCN channel activation. Namely, we observed that at RMP values above −66 mV (i.e., in SCI_DP_ neurons), most HCN channels were closed (Figure 2E). In consequence, the closed HCN channels no longer controlled the intrinsic excitability of the axotomized SCI_DP_ neurons adequately. In contrast, in neurons that were still sufficiently polarized, i.e., SCI_HP_ neurons, as well as uninjured neurons, HCN channels were active, and they did indeed control the RMP, as well as the intrinsic neuronal excitability (Figure 5). Accordingly, the selective blocker ZD7288 modulated the activity of axotomized SCI_HP_ neurons, hyperpolarizing their RMP and decreasing their intrinsic excitability. For the opposite situation, ZD7288 failed to hyperpolarize the RMP and failed to decrease the hyperexcitability of SCI_DP_ neurons. Hence, the “rescuing” effect of ZD7288 occurred only in the healthy and in the least impaired M1LV neurons, and it did not occur in the most impaired neurons.

This work demonstrates that HCN channels are fully functional after M1LV neuron axotomy. Therefore, it is tempting to question whether the pharmacological control of HCN channels can be exploited to modulate the excitability of axotomized M1LV neurons that are sufficiently polarized. Additionally, HCN channel blockage could even modulate the most depolarized neurons if occasional channel activation occurred in vivo, e.g., due to membrane potential oscillations [32]. Answering such a question is not trivial, because of the dual role of HCN channels in the control of neuronal excitability, involving both pro-excitatory effects based on the channel ability to control the RMP and pro-inhibitory effects based on the channel ability to control the R_in_ [19]. As a result of this dual control, both the increase and the decrease of HCN channel function may eventually lead to neuronal hyperexcitability [33,34,35,36,37]. This hinders a straightforward prediction of the consequences of chronic pharmacological HCN channel blockage after SCI. Addressing the matter directly, our data revealed that, in healthy M1LV neurons, a larger I_h_ was associated with a more depolarized RMP. In turn, the blockage of HCN channels caused RMP hyperpolarization and decreased intrinsic neuronal excitability. Thus, the predominant activity of HCN channels in our system was pro-excitatory and RMP-mediated. 

Since HCN channel blockage had the same effects in uninjured and in SCI_HP_ neurons, we can conclude that HCN channels retain their physiological role in most of the axotomized neurons. Therefore, it would be tempting to predict that blocking or reducing HCN-mediated currents after SCI will be helpful to contain cortical hyperexcitability. However, such a prediction would be simplistic. Indeed, we need to account for the fact that HCN channel blockage caused an increase in R_in_ due to the reduction of ionic conductance across the cell membrane [19]. As we previously described for M1LV neurons, R_in_ and rheobase current amplitude are inversely related, and their relation approximates a hyperbole [38]. Therefore, the R_in_-mediated effects of HCN channel blockage are pro-excitatory. In our system, the pro-excitatory effects of the HCN channel blocker were subtler than the pro-inhibitory effects (RMP-mediated) and unmasked only after setting the baseline voltage to −70 mV. Nevertheless, such minor pro-excitatory effects may be enhanced by combining HCN channel blockage with network disinhibition, which would largely increase the M1LV output evoked by cortical neurotransmission [39]. Cortical disinhibition often follows SCI [5] and is subject to phasic fluctuation over extended periods after injury [40]. Thus, as the motor cortex enters and exits phases of disinhibition after SCI, the effects of reduced HCN channel activation may vary, and the effects of HCN channel blockage may change over time. 

From the perspective of clinical applications, even after assuming consistent effects of pharmacological HCN channel blockage within each individual phase of recovery, it is necessary to consider that the channel blockage might yield both advantages and disadvantages. As a potential risk, transient disinhibition facilitates network remodeling and recovery [20,41,42], but it comes at the price of aggravating symptoms such as pain, spasticity, and poor motor coordination [3,4,43,44]. Thus, appropriate modulation of cortical excitability will rely on understanding how changes in intrinsic neuronal excitability and phasic disinhibition interact over time after SCI and teasing apart beneficial and detrimental effects according to individual timeline of recovery. On the other side, reducing the depolarization and hyperactivity of motor neurons may help to preserve corticospinal neurons in a healthier state. In fact, our data indicated that even in the SCI_HP_ population, in which chronic depolarization was not an issue, there was an augmented excitability in comparison to uninjured neurons (Figure 4). Thus, hyperexcitability is common to SCI_HP_ neurons and SCI_DP_ neurons. Additionally, a depolarized RMP aggravated the conditions of SCI_DP_. 

Chronic depolarization and excessive excitability can be caused by multiple factors, including altered cationic conductances, cellular stress, and anomalous intracellular calcium buffering [45,46,47,48,49]. Any of these may apply to M1LV neurons heterogeneously. While the molecular components remain to be determined, it will be helpful to question whether different pathological pathways tease apart SCI_HP_ and SCI_DP_ neurons, or whether the two groups represent different timepoints in the course of the same process. Especially in case of the latter possibility, preventing or reducing the excessive depolarization of SCI_DP_ neurons would be helpful to protect them from cellular damage and excitotoxicity, which may cause the reported slow and progressive loss of M1LV neurons after SCI [10].

Preclinical and clinical studies demonstrate that the modulation of cortical excitability after SCI is relevant for both the spontaneous and assisted regeneration of axons near the point of injury [12,13,50,51,52,53]. In relation to these matters, our work sheds light on corticospinal excitability after SCI as a basis to control network plasticity to rebuild circuits with functional properties better suited to improving the corticospinal function after injury [5].

## 4. Materials and Methods

### 4.1. Surgery

Experiments were performed in agreement with the “Directive 2010/63/EU of the European Parliament and of the Council of 22 September 2010 on the protection of animals used for scientific purposes” and were approved by Austrian Federal care authorities: protocol number 2020-0.073.987.

Surgeries were performed on female Fisher-344 rats of 12 weeks of age (Charles Rivers Laboratory, Sulzfeld, Germany) under general anesthesia by inhalation of isoflurane/oxygen mix, using a small animal anesthesia unit (SomnoSuite, TSE Systems GmbH, Hamburg, Germany). Analgesia was provided by a subcutaneous injection of buprenorphine (0.03 mg/kg bodyweight), at least 45 min before surgery. Wire-knife transection of the dorsal corticospinal tract at the spinal cord level C4 and postoperative care were performed as previously described [14]. Briefly, after skin incision and exposure of the dorsal spine, laminectomy was carried out at cervical level C4, and the dorsal corticospinal tract (CST) was transected with a blunt tungsten wire-knife, with an incision of approximately 2.5 mm in width and 1.1 mm in depth. Upon CST transection, 2 µL of fluorogold (Fisher Scientific; 4% *w/v* in NaCl 0.9% solution) was injected in the lesion site to label transected axons. Afterward, the paravertebral muscles were sutured, and the skin was closed with staples. Following surgery, rats received enrofloxacin (10 mg/kg bodyweight) and meloxicam (2 mg/kg bodyweight) daily for five days and buprenorphine (0.03 mg/kg bodyweight) for two days twice a day.

### 4.2. Electrophysiology

Before electrophysiological analysis, rats were briefly sedated with isoflurane and decapitated. Brains were dissected while submerged in chilled artificial cerebrospinal fluid (ACSF). Chilled high-sucrose ACSF was used for slice preparation and contained (in mM): 206 sucrose, 25 NaCO_3_, 25 glucose, 1.0 CaCl_2_, 3.0 MgCl_2_, 2.5 KCl, and 1.25 NaH_2_PO_4_; osmolarity = 309 mOsm [54]. Coronal sections were sliced with a Leica VT1200s microtome at a thickness of 250 µm while submerged in chilled ACSF. After dissection, acute slices were stored in ACSF at room temperature. The ACSF used for acute slice storage, as well as for patch-clamp measurements, contained (in mM): 134 NaCl, 26 NaHCO_3_, 25 glucose, 2 CaCl_2_, 1 MgCl_2_, 2.4 KCl, and 1.25 NaH_2_PO_4_; pH was balanced to 7.4, using a mix of CO_2_/O_2_ (95/5%); osmolarity = 315 mOsm. For the blockage of I_h_ currents, the HCN channel blocker ZD7288 (100 µM, Tocris, Bristol, UK) was added to the ACSF and applied by perfusion into the recording chamber during the patch clamp experiments. The experimental setup consisted of an Olympus upright microscope equipped with motorized micromanipulators and stage (Scientifica, Uckfield, UK). The recording chamber contained an ACSF volume of 0.5–1.0 mL, exchanged at a flow rate > 1.0 mL/min. Patch pipettes had a resistance of 5–6.5 MΩ and were allowed to achieve a comparable *R*_S_ in different groups (29 ± 8 MΩ in SCI and 28 ± 10 MΩ in uninjured neurons; *p* = 0.68). The intracellular pipette solution contained (in mM): 135 K-gluconate, 4 KCl, 10 HEPES, 10 Na-phosphocreatine, 4 ATP-Mg, 0.3 GTP-Na [54], and biocytin (1 mg/mL). The pH was adjusted to 7.25; osmolarity = 300 mOsm. Osmolarity was measured with a Vapro (Wescor, Logan, UT, USA) osmometer. 

In acute slices, principal neurons in the lower part of M1LV were readily identified according to the morphology of their soma. During patch clamp experiments, cells were dialyzed with biocytin. Axotomized neurons were labeled by Fluorogold (FG), which is a retrograde tracer injected at vertebral level immediately after the wire-knife lesion. FG is taken up by axotomized axons, resulting in consistent labelling of cortical motor neurons upon CST transection [10]. After the patch-clamp experiments, tissue fixation and fluorophore-conjugated streptavidin staining allowed us to identify axotomized neurons in rats that had undergone SCI. When biocytin and FG localization could not be accurately assessed (e.g., due to damage or removal of the soma after patch-pipette retraction), electrophysiological data were not analyzed. Electrophysiological data were acquired with a HEKA amplifier (HEKA, Lambrecht, Germany) at a sampling rate of 10 KHz, filtered at 2 KHz, and analyzed with FitMaster (HEKA), Origin (OriginLab, Northampton, MA, USA) and Prism 9 (GraphPad, San Diego, CA, USA). Rheobase current was determined with current clamp protocols, consisting of consecutive 500 ms long hyperpolarizing and depolarizing steps from resting membrane potential (RMP). Hyperpolarizing steps starting at −20 pA, adding 5 pA to each consecutive step, until rheobase was reached. To determine the relation between input current and action potential (AP) frequency, larger steps were used (20 pA), starting at −200 pA and up to 250–300 pA. In between, the voltage was maintained at RMP for 500 ms. Voltage-clamp protocols used to determine the current–voltage relation of voltage-gated currents consisted of hyperpolarizing steps (5 mV) of 2 s, from −140 mV to −60 mV. Between steps, voltage was held at a −60 mV for 4 s. R_in_ was calculated as ratio between voltage and membrane current (*I*_mem_) elicited by a −5 mV hyperpolarizing voltage step applied at −70 mV. The I_h_ was determined as the slowly activating current component elicited at a voltage of −140 mV and selectively blocked by ZD7288 (Figure 2A). I_h_ fractional activation was determined as normalized amplitude of tail currents selectively blocked by ZD7288 (Figure 2A,B).

### 4.3. Histology

After patch-clamp experiments, acute slices were fixed for 1 h in a 0.1 M phosphate-buffered paraformaldehyde 4% solution (pH 7.4). Subsequently, slices were washed repeatedly in 0.1 M phosphate buffer solution before being incubated with blocking-buffer for 60 min (1% *w/v* BSA, 0.2% fish skin gelatin *v/v*, and 0.1% *v/v* Triton X-100 in PBS). Primary antibody incubation was carried out overnight at 4 °C. Primary antibodies were guinea-pig anti-NeuN (1:750, MerkMillipore, Burlington, MA, USA) and rabbit anti-Fluorogold (1:750, Fluorochrome, Denver, CO, USA). After primary antibody incubation, slices were washed in PBS repeatedly prior to incubation for 90 min with secondary antibodies. Secondary antibodies were donkey-anti rabbit Alexa 405 (1:500, Abcam, Cambridge, UK) and donkey anti-guinea pig Alexa 568 (1:1000, ThermoFisher Scientific). Streptavidin staining (Streptavidin-Alexa-488, ThermoFisher Scientific, Waltham, MA, USA) was carried out, along with secondary antibody incubation to outline neurons dialyzed with biocytin during the patch clamp experiments. For preservation of immunofluorescence, slices were mounted on coverslips by using ProLong Gold Antifade (ThermoFisher Scientific, Waltham, MA, USA). Fluorescence images were acquired using a confocal microscope LSM 710; laser lines 405 nm, 488 nm, and 568 nm; 20× dry objective (NA = 0.8); and the software ZEN (Carl Zeiss Microscopy, Oberkochen, Germany).

### 4.4. Statistics

Data were analyzed with GraphPad Prism 8. Significance was determined with the Mann–Whitney test or with an unpaired t test, according to parametric or non-parametric sample distribution. Significance in multiple comparisons was calculated with one–way ANOVA and Holm–Sidak’s multiple comparison test or Kruskal–Wallis and Dunn’s multiple comparison test, according to parametric or non-parametric sample distribution. Data are represented as mean ± standard deviation, and statistically significant differences were assumed for *p*-values < 0.05. 

## Figures and Tables

**Figure 1 ijms-24-04715-f001:**
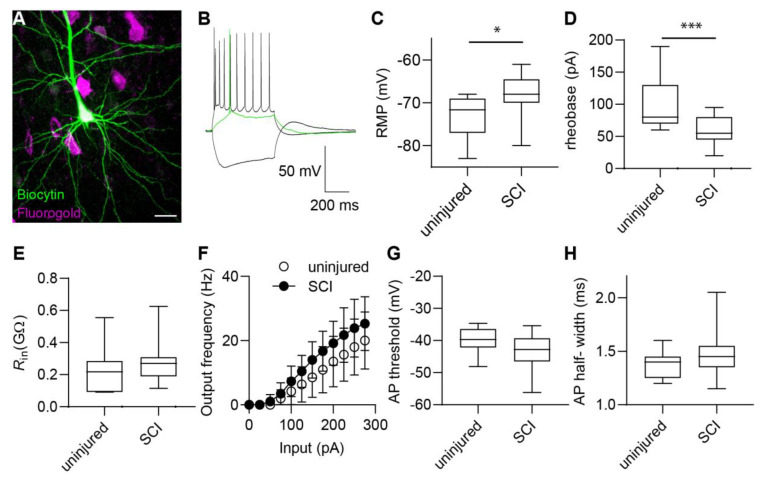
Increased excitability in M1LV neurons after SCI. (**A**) Morphology of M1LV neuron displayed by biocytin dialysis during patch-clamp experiments and streptavidin staining post-fixation (green). Fluorogold injected in the spinal cord at the time of injury (magenta) highlights axotomized M1LV neurons by retrograde labeling (co-labeling appears in white). (**B**) Typical AP firing pattern elicited in M1LV neurons upon chronic depolarization. Green trace highlights AP firing at rheobase. (**C**) The resting membrane potential (RMP) in M1LV neurons is depolarized significantly after SCI. (**D**) Rheobase in M1LV neurons is decreased significantly after SCI. (**E**–**H**) Input resistance (R_in_) (**E**), relation of input (current) amplitude, and output (AP) frequency (**F**), AP threshold (**G**), and AP half-width (**H**) are not significantly altered in M1LV neurons after SCI. * *p* < 0.05; *** *p* < 0.001; scale bar = 20 µm.

**Figure 2 ijms-24-04715-f002:**
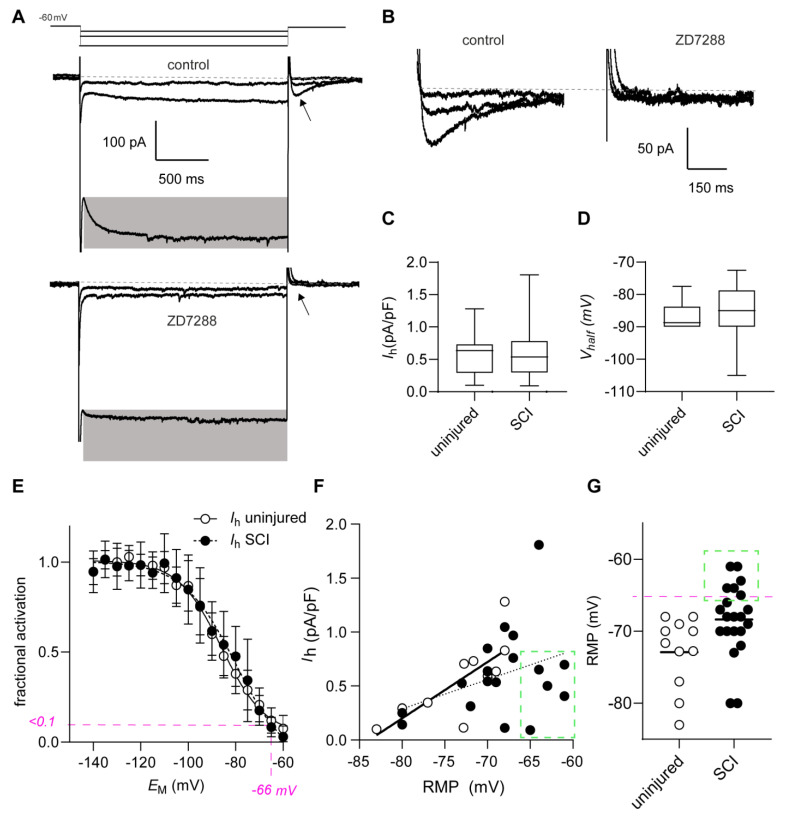
Relation between depolarization and HCN-channel-mediated currents in M1LV neurons. (**A**) Hyperpolarizing voltage steps from a holding potential of −60 mV elicit slowly activating inward currents (highlighted in gray), which are blocked by ZD7288. Currents selected for graphic display were elicited by hyperpolarizing sweeps to −135 mV, −75 mV, and −65 mV. (**B**) HCN-channel-mediated tail currents elicited by membrane depolarization (−60 mV) from hyperpolarized potentials (highlighted in (**A**) by arrowhead). Tail currents are blocked by ZD7288. (**C**) The maximal current (*I*_h_) density in M1LV neurons does not change significantly after SCI. (**D**) Voltage of half-maximal activation and (**E**) I_h_ fractional activation are not affected by SCI. Pink dotted line highlights voltage of −66 mV, at which I_h_ activation is negligible (<0.1). (**F**) There is a significant correlation between the RMP and the maximal I_h_ current amplitude in uninjured neurons (solid line shows linear regression). Correlation is not significant after SCI (dotted line shows linear regression). Green inset highlights depolarized RMP and small I_h_ amplitude (**G**). The RMP of some axotomized neurons is considerably depolarized after SCI (≥−66 mV), whereas the I_h_ activation at RMP in these cells is negligible ((**E**) pink dotted line), and the maximal I_h_ current amplitude is low (**F**).

**Figure 3 ijms-24-04715-f003:**
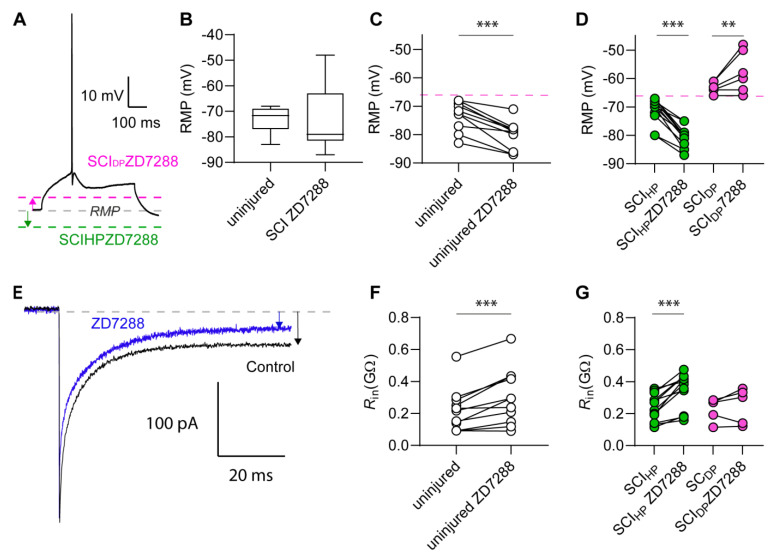
Heterogeneous effects of HCN channel blockage in M1LV neurons after SCI. (**A**) The RMP of uninjured and axotomized neurons is monitored in current clamp experiments at baseline voltage before the application of current steps that elicit AP firing. (**B**) Application of ZD7288 has a rescuing effect on the RMP of axotomized neurons, which is no longer significantly different from the RMP of uninjured neurons. (**C**) ZD7288 causes significant RMP hyperpolarization in uninjured neurons. (**D**) ZD7288 causes RMP hyperpolarization in axotomized neurons (SCI_HP_), of which the RMP was originally <−66 mV (red line). Conversely, ZD7288 causes RMP depolarization in neurons (SCI_HP_) which originally had an RMP of ≥−66 mV. (**E**) Representative current elicited by a −5 mV hyperpolarizing voltage step applied at −70 mV. Arrows indicate membrane current (I_mem_). Note the smaller I_mem_ highlighted by the blue arrow corresponding to increased R_in_ in the presence of ZD7288. (**F**,**G**) Changes in R_in_ occur in axotomized (**F**) and uninjured neurons (**G**) alike. ** *p* < 0.01; *** *p* < 0.001.

**Figure 4 ijms-24-04715-f004:**
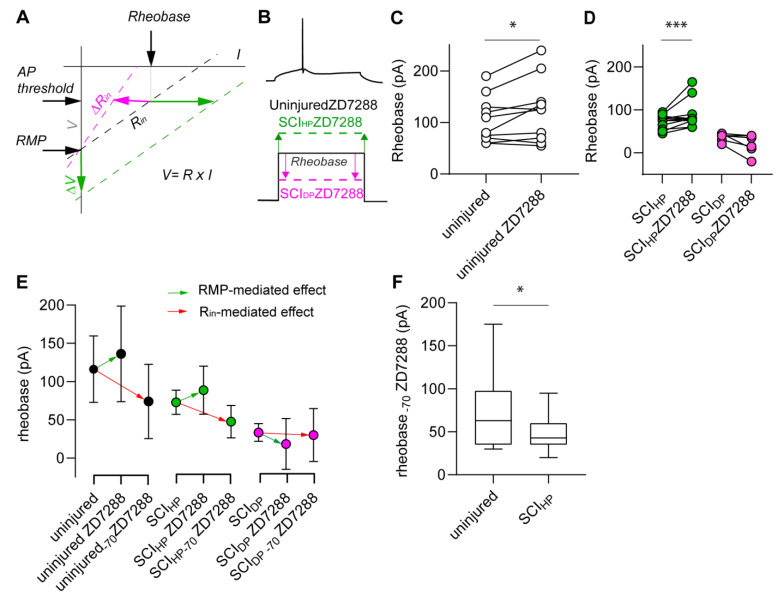
ZD7288 has RMP-mediated effects and R_in_-mediated effects that influence the rheobase of M1LV neurons. (**A**) The linear relation between voltage and current (V = R × I) explains why RMP hyperpolarization contributes to increased rheobase per se (green arrows), while reduced R_in_ has the opposite effect per se (red arrows). Note: Upon ZD7288 application, both effects are combined. (**B**–**D**) Current clamp experiments reveal different effects of ZD7288 on the rheobase of uninjured neurons, SCI_HP_ neurons, and SCI_DP_ neurons. (**C**) ZD7288 causes a significant increase of rheobase in uninjured neurons. (**D**) Increased rheobase occurs in SCI_HP_, and a slight decrease in rheobase occurs in SCI_DP_. (**E**) RMP-mediated effects of ZD7288 on the rheobase (green arrows) are prevented by equalizing the baseline voltage to −70 mV (close to RMP of untreated neurons). Thus, the R_in_-mediated effects of ZD7288 are highlighted by a strong decrease in rheobase (red arrow) in uninjured and axotomized neurons alike, with the exception of SCI_DP_. (**F**) In the presence of ZD7288 and from an equalized baseline voltage of −70 mV, the rheobase of SCI_DP_ neurons is significantly lower than that of uninjured neurons. * *p* < 0.05; *** *p* < 0.0001.

**Figure 5 ijms-24-04715-f005:**
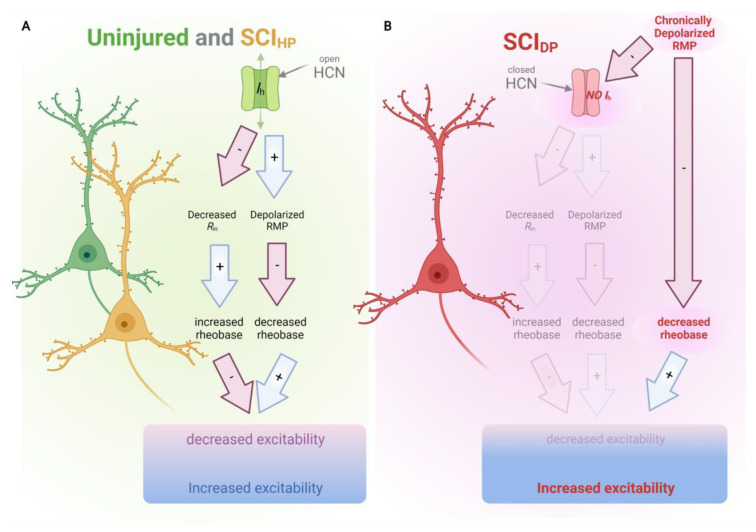
Model of physiological consequences of reduced HCN activation in SCI_DP_. (**A**) In uninjured neurons, and in axotomized SCI_HP_ neurons, physiological levels of HCN channel activation elicit I_h_ currents, which decrease the R_in_ (which, in turn, increases the rheobase) and depolarize the RMP (which, in turn, decreases the rheobase). Thereby physiological HCN channel activation mediates both positive and negative regulation of neuronal excitability, and the two control components are in balance. (**B**) In axotomized SCI_DP_ neurons, chronic RMP depolarization causes decreased HCN channel activation and a consequent lack of HCN-mediated regulation of neuronal excitability. Additionally, chronic depolarization has per se an influence on the rheobase, contributing to increased excitability directly. Thus, the components that control neuronal excitability are imbalanced. Image created with Biorender.com (accessed on 16 January 2023; agreement number XB24WB0WTV).

**Table 1 ijms-24-04715-t001:** Physiological properties of M1LV neurons.

	Uninjured (*n* = 11)	SCI (*n* = 21)	*p*-Value
RMP (mV)	−72.9 ± 5.0	−68.4 ± 5.1	0.0108 ^1^
Rheobase (pA)	59.3 ± 21.2	102.3 ± 43.7	<0.001 ^1^
R_in_ (GΩ)	0.22 ± 0.14	0.26 ± 0.12	0.1167 ^2^
AP threshold (mV)	−40.2 ± 4.1	−43.4 ± 5.3	0.1100 ^1^
AP half-width (ms)	1.37 ± 0.24	1.47 ± 0.21	0.1000 ^2^
I_h_ (pA/pF)	0.6 ± 0.3	0.6 ± 0.4	0.4736 ^2^
C_M_ (pF)	258.5 ± 130.5	191.9 ± 71.3	0.1390 ^1^
I_h_ V_half_	−86.9 ± 4.6	−85.3 ± 9.6	0.1506 ^2^
RMP/I_h_ correlation (R)	0.76	0.36	0.0067 (uninjured) 0.1429 (SCI)

^1^ Parametric test. ^2^ Non-parametric test. Membrane capacitance (C_M_); voltage of I_h_ half-maximal activation (V_half_).

**Table 2 ijms-24-04715-t002:** Effects of ZD7288 on M1LV neurons after SCI.

	Uninjured (*n* = 11)	Uninjured ZD7288 (*n* = 11)	Adjusted *p*-Value
RMP (mV)	−72.9 ± 5.0	−80.1 ± 4.9	<0.001 ^2^
R_in_ (GΩ)	0.22 ± 0.14	0.30 ± 0.17	<0.001 ^2^
Rheobase (pA)	102.3 ± 43.7	117.7 ± 61.5	0.0300 ^1^

^1^ Parametric test. ^2^ Non-parametric test.

**Table 3 ijms-24-04715-t003:** Effects of ZD7288 on two groups of axotomized M1LV neurons after SCI.

	SCI_HP_ (*n* = 12)	SCI_HP_ ZD7288	SCI_DP_ (*n* = 6)	SCI_DP_ ZD7288	Adjusted *p*-Value
RMP (mV)	−71.2 ± 4.5	−80.7 ± 3.7	−63.1 ± 1.9	−57.7 ± 7.3	<0.0001 ^1a^ 0.0051 ^1b^
R_in_ (GΩ)	0.24 ± 0.08	0.35 ± 0.11	0.23 ± 0.07	0.25 ± 0.11	<0.0001 ^1a^ 0.3868 ^1b^
Rheobase (pA)	72.9 ± 15.9	88.7 ± 31.4	33.3 ± 11.5	20.0 ± 34.6	0.0007 ^1a^ 0.3237 ^1b^

^1a^ Mixed-effect analysis multiple comparisons (SCI_HP_ vs. SCI_HP_ ZD7288). ^1b^ Mixed-effect analysis multiple comparisons (SCI_DP_ vs. SCI_DP_ ZD7288).

**Table 4 ijms-24-04715-t004:** Rheobase of M1LV neurons measured from equalized membrane voltage (−70 mV), in presence of ZD7288.

	Uninjured (*n* = 8)	SCI_HP _ (*n* = 12)	SCI_DP_ (*n* = 3)	*p*-Value
Rheobase_-70_ (pA)	74.04 ± 48.5	47.6 ± 21.3	30.0 ± 34.6	0.0300 ^1^ n.a. ^2^

^1^ Parametric test: uninjured vs. SCI_HP_; ^2^ n.a., not applicable: uninjured vs. SCI_DP._

## Data Availability

Supporting data available: www.researchgate.net/publication/368834149_M1LVSCI-Ih-data.

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
