# Peer review of "Depolarization and Hyperexcitability of Cortical Motor Neurons after Spinal Cord Injury Associates with Reduced HCN Channel Activity"

_ijms, 2023, doi:10.3390/ijms24054715_

Round 1

Reviewer 1 Report

In the present study, the authors made tight-seal whole-cell recordings from principal neurons of the primary motor cortex layer V (M1LV) in brain slice preparation. They compared the membrane properties of M1LV neuros between spinal cord-injured rats and naïve rats. They also studied the effects of ZD7288, an antagonist of hyperpolarization cyclic nucleotide-gated (HCN) channels. They found that some axotomized M1LV neurons became excessively depolarized, and the HCN channels were less active in these neurons. 

The topic is interesting and worth investigating. Such a study may provide the necessary information to further the understanding of spinal cord injury pathophysiology. However, I have the following minor concerns: 

1.     The authors described the existence of two types of cortico-spinal neurons—one depolarizes (SCIDP neurons) in response to spinal cord injury, and the other hyperpolarizes (SCIHP neurons).

Please explain what cellular and/or molecular mechanisms underlie these different responses to spinal cord injury.

Also, please describe the physiological and pathophysiological significance surrounding the existence of these two neuron types.

2.     Although I am not a native speaker of English, there are some unusual expressions and grammatical errors that make the manuscript difficult to read. 

Please make the manuscript more readable for non-native English speakers. 

3. Figure 3G is incorrectly labeled as “SCDP” and should be labeled as “SCIDP.”

Author Response

Reviewer comment 1. 

The authors described the existence of two types of cortico-spinal neurons—one depolarizes (SCIDP neurons) in response to spinal cord injury, and the other hyperpolarizes (SCIHP neurons).

Please explain what cellular and/or molecular mechanisms underlie these different responses to spinal cord injury.

Also, please describe the physiological and pathophysiological significance surrounding the existence of these two neuron types.

Reply 1.

We thank the reviewer for the positive criticism. As a small remark to this comment: SCIDP neurons were depolarized (RMP above -65 mV) in response to SCI.

SCIHP neurons did not hyperpolarize in response to SCI: their RMP was relatively hyperpolarized in respect to SCIDP, but the RMP of SCIHP neurons was also closely comparable to that uninjured neurons (see also Table 2 and Table 3). Indeed, SCIHP neurons were mostly comparable to uninjured neurons in every aspect of their physiology, except for a low rhehobase-70 in presence of ZD7288 (fig. 4 F). We hope that our thorough work of revision has now clarified the less understandable parts of the text.

About the reason for the existence of two populations and the mechanisms driving such pathophysiology: we originally hypothesized that the root cause of the functional alterations after SCI would be a dysfunctional regulation of HCN channels (this was the primary hypothesis of our work). We clearly show that this is not the case. Accommodating your request, we now expanded part of our discussion on this matter in the manuscript, along with adding some speculation about the reasons for the existence of the two populations and the possible pathophysiological mechanisms contributing to the neuronal dysfunction after axotomy.

Additionally, we would like to share the following thoughts.

We cannot exclude that the pathophysiology of SCIHP and SCIDP neurons might be driven by different mechanisms (which we did not explore) and/or associated to a different degree of vulnerability amongst the two groups of neurons.

However, we suspect that the different states of depolarization / hyperexcitability in M1LV represent cells at a different individual time point within the same dynamic process of degeneration. Possibly, the time course varies slightly between individuals and/or between different neurons. If so, testing the system at multiple time points and with models of different SCI severity could change the proportion between SCIHP and SCIDP neurons.

There is a possible complication in investigating single-cell pathophysiology over time: Hains et al. 2003 reports loss of M1LV neurons two weeks after SCI, while one week after injury such loss is still negligible. If the SCIDP neurons are those that dye first (e.g. due to excitotoxicity) we might detect fewer or no SCIDP neuron at later time points… unless SCIHP depolarize over time.

Hearing the opinion of different researchers in the field of SCI, we understand that the publication by Hains is somehow controversial. Because of this, and due to the speculative nature of our ideas, we maintained the discussion about pathophysiological mechanisms as restrained as possible. Rather, we are implementing other paradigms of analysis in our ongoing (unpublished) work, to account for potential losses of M1LV neurons from one week after SCI onwards.

Reviewer comment 2.     

Although I am not a native speaker of English, there are some unusual expressions and grammatical errors that make the manuscript difficult to read. 

Please make the manuscript more readable for non-native English speakers. 

Reply 2.

Heeding your advice, we corrected the text extensively and simplified the manuscript to improve readability. The text has been revised by the authors and by two native English speakers. We hope you will find the new version improved suitably.

Reviewer comment 3.

Figure 3G is incorrectly labeled as “SCDP” and should be labeled as “SCIDP.”

Reply 3.

Thank you for spotting this mistake! We corrected the erroneous label and made some additional aesthetic improvements to the figures (colour code), to make them friendly for colour-blind persons.

Reviewer 2 Report

In this work, the role of HCN channel activity injury is studied. First, the authors describe the increase in excitability in M1LV neurons after spinal cord injury and the relationship with the HCN channels. Interestingly and unexpectedly, the authors show different effects in response to HCN channel blocker on uninjured vs. injured neurons. Finally, the authors propose a model of the possible consequences of reducing HCN activation after injury. This work is of great interest to understand the mechanisms by which the excitability of injured neurons is altered. The modulation of the excitability of neurons after an injury could be a key target to promote regeneration after injury.

Author Response

Comment:

In this work, the role of HCN channel activity injury is studied. First, the authors describe the increase in excitability in M1LV neurons after spinal cord injury and the relationship with the HCN channels. Interestingly and unexpectedly, the authors show different effects in response to HCN channel blocker on uninjured vs. injured neurons. Finally, the authors propose a model of the possible consequences of reducing HCN activation after injury. This work is of great interest to understand the mechanisms by which the excitability of injured neurons is altered. The modulation of the excitability of neurons after an injury could be a key target to promote regeneration after injury.

Reply:

We thank the reviewer for the positive feedback and we are happy that the manuscript met their approval. We share interest and excitement about understanding the mechanisms of cortical pathophysiology after SCI and we look forward to endeavouring in our efforts to control neuronal excitability and helping the recovery and regeneration.